# Supply-side options to reduce land requirements of fully renewable electricity in Europe

Tim Tröndle[1,2]*

**1** Institute for Advanced Sustainability Studies, Potsdam, Germany, **2** Institute for Environmental Decisions, ETH Zürich, Zürich, Switzerland

* tim.troendle@iass-potsdam.de

## Abstract

Renewable electricity can fully decarbonise the European electricity supply, but large land requirements may cause land-use conflicts. Using a dynamic model that captures renewable fluctuations, I explore the relationship between land requirements and total system cost of different supply-side options in the future. Cost-minimal fully renewable electricity requires some 97,000 km$^2$ (2% of total) land for solar and wind power installations, roughly the size of Portugal, and includes large shares of onshore wind. Replacing onshore wind with offshore wind, utility-scale PV, or rooftop PV reduces land requirements drastically with only small cost penalties. Moving wind power offshore is most cost-effective and reduces land requirements by 50% for a cost penalty of only 5%. Wind power can alternatively be replaced by photovoltaics, leading to a cost penalty of 10% for the same effect. My research shows that fully renewable electricity supply can be designed with very different physical appearances and impacts on landscapes and the population, but at similar cost.

## 1. Introduction

Europe has the potential to generate all its electricity from renewable sources [1–3]. The potential provides a possibility to decarbonise the European electricity system, which is a necessary step to reach the European Commission's goal of becoming a climate-neutral economy by 2050 [4]. Compared to the predominant forms of electricity supply based on fossil and nuclear fuels, land requirements of renewable electricity are high [5–8], however. A transition to renewable electricity will therefore increase the total land requirements of electricity supply and it may even do so by orders of magnitude.

While renewable electricity is an indispensable option to mitigate global climate change, its high land requirements have the potential to cause conflicts locally where it is built. This is for three reasons. First, it may compete with other uses of land. Of the main two current technologies of renewable electricity, photovoltaics and wind turbines, only the latter allows for limited dual use of land: for technical reasons, spacing between turbines is large and, as a result, that land can be used for agriculture [7]. Second, renewable generation infrastructure has the potential to economically devalue land on which it is built, and also neighbouring land. There

**Data Availability Statement:** The applied model of the electricity system and all result data are available from Zenodo doi:10.5281/zenodo.3707812.

**Funding:** The work of T.T. was supported by a European Research Council grant (TRIPOD, grant

agreement number 715132). http://erc.europa.eu
The funders had no role in study design, data
collection and analysis, decision to publish, or
preparation of the manuscript.

**Competing interests:** The author has declared that
no competing interests exist.

is conflicting evidence whether wind power in sight of properties impacts property values, and
while the majority of studies do not find statistically significant impacts, some others find
losses in property values of up to 15% [9]. Third, wind [10] and solar power [11] are sometimes
perceived as negatively impacting the landscape, depending on place attachment and the aes-
thetics of the previously undisturbed landscape [10,12,13], and location and density of struc-
tures [10].

While the acceptance of the energy transition is generally high and the majority of the pop-
ulation does not feel disturbed by wind and solar installations [14,15], local opposition has hin-
dered and delayed local renewable electricity projects in the past [16,17]. Opposition may
continue in the future, considering the large expansion of impacted land area moving towards
fully decarbonised electricity supply [15]. This led some authors to the conclusion that fully
renewable electricity–while being theoretically possible–will not be feasible in Europe [5,7].

Strategies to mitigate negative impacts associated with the land requirements of renewable
electricity include location and placement of generation infrastructure [10] and technology
choice to reduce land requirements [18]. If proven effective, these strategies can not only
reduce negative side-effects, but also increase the feasibility of electricity systems with large
shares of renewable electricity by reducing opposition.

In this article, I explore the relationship between land requirements and total system cost of
fully renewable electricity systems in Europe with different supply sides. Renewable supply
technologies have vastly different land requirements, with two orders of magnitude between
the land requirements of bioenergy with dedicated farming for crops, the technology with the
largest land requirements, and solar electricity, which has the lowest [7,8]. Previous research
shows that fully renewable electricity supply in Europe is possible in many different ways, with
very different shares of supply technologies, and that cost differences between supply options
can be low if designed right [19–21]. However, while several studies have assessed cost [1,22–
26] and land requirements [1,3,27,28] of supply technologies and entire electricity systems,
only one study has assessed the relationship between the two on the system level [18]. The sys-
tem perspective is central to renewable electricity systems as it takes into account not only the
supply side but also technologies to handle fluctuations of the supply side. In their case study
of Alberta, Canada, the authors find higher total system cost for lower land requirements even
though they allow for large amounts of electricity from non-renewable sources. No study has
assessed the relationship between land requirements and cost on the system level using only
renewable resources in Europe. I address the relationship in this study by determining cost-
effective ways to reduce land requirements of fully renewable, future electricity systems in
Europe through supply technology choice.

To find cost-effective ways to reduce land requirements, I do the following: I use a nation-
ally resolved, dynamic model of the European electricity system to determine total system cost
and land requirements of fully renewable supply. I find that, while there is a trade-off between
cost and land requirements, systems with vastly different requirements for land can be built
with small cost penalties.

## 2. Data and methods

To identify cost-effective ways to reduce land requirements by supply technology choice, I
generate total system cost and total land requirement data for fully renewable electricity supply
in Europe with different shares of supply technologies using a model-based approach. I gener-
ate the data in two steps. First, I generate cost-minimised system designs for 286 different
shares of supply technologies. Second, I determine total system cost and land requirements for
each system design. Using the Monte Carlo method respecting uncertainty of technology

parameters, I create 100,000 samples for each system design. In total, I end up with ~29 million observations of pairs of cost and land requirements. All data procedures and all analysis steps are publicly available [29] as a Snakemake workflow [30].

The following subsections explain all data generation and analysis steps in more detail.

## 2.1 Capacity shares of supply technologies

To understand how supply technology choice can mitigate land requirements of renewable electricity in Europe, I enforce different capacity shares of technologies in the system designs that I am analysing. The geographic scope of this study includes most countries with member organisations in the entso-e: EU-27 without Cyprus, the United Kingdom, Switzerland, Norway, and Western Balkan countries. I focus on four dominant wind and solar supply technologies: on- and offshore wind, and ground- and roof-mounted photovoltaics (PV). I analyse all 286 possible combinations based on ten different shares per technology, from 0% through 10% and 20% up to 100% (Fig 1). The shares are applied to the European level, but also to the national level, meaning that each country in Europe has to meet shares individually. Furthermore, I assume each country to be net self-sufficient, generating enough electricity annually to fulfil its domestic electricity demand but able to trade with other countries to balance renewable fluctuations.

Other than these four supply technologies, the system designs furthermore contain hydroelectricity with and without reservoirs and bioenergy, all of which can generate renewable electricity to a limited extent as well. While I do not restrict their capacity shares, they are both restricted by their generation potential: hydroelectricity is limited to the amount that can be generated using current capacities, and bioenergy is limited to the amount of bioenergy that can be generated from residuals (see System design model). Both contribute to electricity supply, but only to minor extents. Hydroelectricity with reservoirs and bioenergy are used to balance renewable fluctuations as well.

Each country in Europe can potentially generate enough electricity from rooftop PV, utility-scale PV, and onshore wind to cover national demand together with limited generation from hydroelectricity and bioenergy (see "System design model" and "Discussion"). Thus, system designs with 100% rooftop PV, 100% utility-scale PV, or 100% onshore wind are possible.

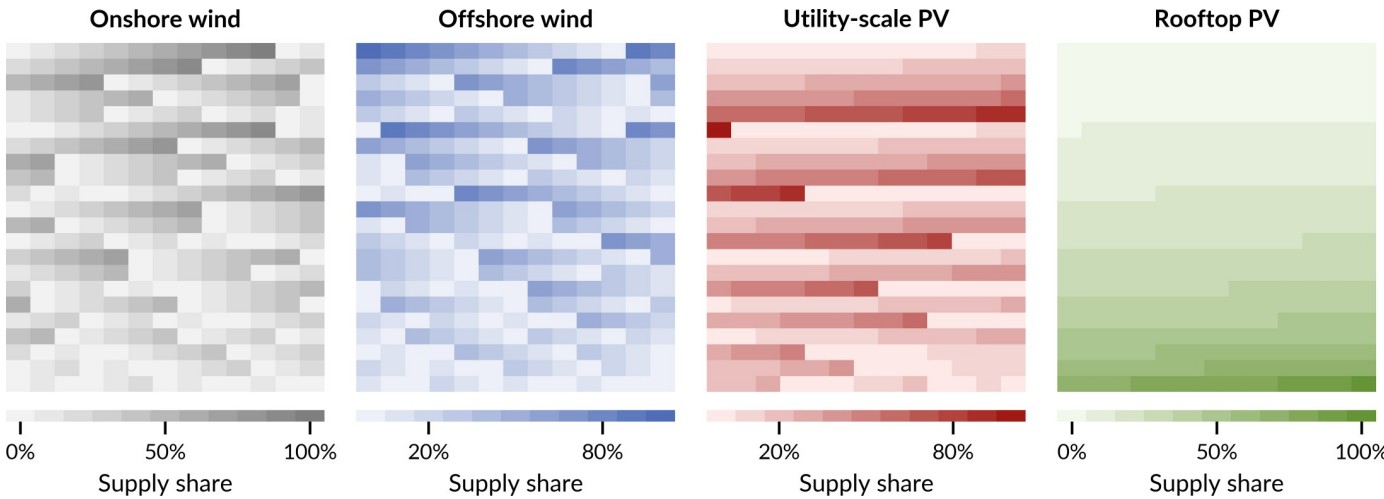

**Fig 1. Capacity shares of all 286 system designs for four supply technologies.** Pixels in each panel represent one system design. System designs are in no particular order. Shares of the same pixel in all panels always add to 100%.

The situation is different for offshore wind: while the generation potential for all Europe is large enough to cover European demand, only countries with shores can build offshore wind farms. In countries without shore I replace offshore wind with onshore wind; i.e. when all countries with shores have to enforce a 40% capacity share of offshore wind, and a 20% capacity share of onshore wind, all countries without shores have to have a 60% capacity share of onshore wind. Countries without shores, or with insufficient offshore potentials are: Austria, Bosnia and Herzegovina, Switzerland, Czech Republic, Hungary, Luxembourg, North Macedonia, Serbia, Slovakia, and Slovenia.

Based on the enforced supply capacity shares, the system design model determines absolute installed capacities in each country for all supply technologies, but also for all storage and cross-border transmission capacities.

## 2.2 System design model

The system design model determines cost-minimal system designs for Europe. The model is a network flow model with the electricity transmission grid at its core [31]. Each country in Europe is modelled as a node on the network and all nodes are connected through the transmission grid. The model has a 3h resolution and simulates one full weather year to cover renewable fluctuations. It is a linear optimisation model that optimises system design and operation simultaneously. The model is implemented using the Calliope model framework [32] and has been used in a former publication. It is described in full detail in ref. [21].

On the supply side, the model contains four main renewable technologies to generate electricity: on- and offshore wind, and utility-scale and rooftop PV. In addition, hydroelectricity with and without reservoirs, and bioenergy can generate electricity. Capacities are limited by their technical potentials, which I derive from ref. [3] for wind and solar power, and from ref. [21] for hydroelectricity. For hydroelectricity, I assume no further expansion from today is possible, and thus its technical potential equates to today's capacity (see Section 1 in S1 File for a discussion of the impact of this decision). For wind and solar power, I allow any capacity up to their technical potential. Generation profiles are based on weather data [33–35] and taken from the same sources as the potentials. Based on the enforced supply capacity shares, the system design model determines cost-minimal capacities of supply technologies.

To balance fluctuating hydro, solar, and wind generation, system designs contain battery storage, hydrogen storage, pumped storage hydroelectricity, and bioenergy. All storage technologies are modelled as single storage tanks with efficiencies, i.e. there is no flow of energy in any other form than electricity. Battery storage can discharge for a maximum of four hours at full power, while hydrogen storage can discharge for at least four hours. Bioenergy is limited by the amount of fuels that can be produced from residuals in each country per year [36], i.e. I do not assume dedicated farming for energy crops. The limited fuel supply and high fuel cost make bioenergy a technology mainly used for balancing, rather than for supplying electricity. Bioenergy and storage capacities other than pumped hydro are not restricted in any way. I assume pumped storage hydroelectricity can not be expanded significantly and it is thus limited to today's capacities [21]. The system design model determines cost-minimal capacities of all balancing technologies.

All supply, balancing, and cross-border transmission capacities have costs: overnight installation costs, annual maintenance costs, and variable costs (Table 1). The technology costs are important determinants of the magnitude of capacities chosen by the system design model. Their values are future projections and assume all technologies are deployed at large scale. In particular, this means that I ignore any forms of transitional effects stemming from technological or financial learning. Together with the technology lifetime, I determine annuities from

**Table 1. Assumptions on installation and maintenance cost of electricity infrastructure.**

| Technology | Overnight cost (€/kW) | Overnight cost (€/kWh) | Annual cost (€/kW/yr) | Variable cost (€ct/kWh) | Lifetime (yr) | Source |
|---|---|---|---|---|---|---|
| Utility-scale PV | 520 | 0 | 8 | 0 | 25 | Ref. [38] Table 7 |
| Rooftop PV | 880 | 0 | 16 | 0 | 25 | Ref. [38] Table 9 |
| Onshore wind | 1100 | 0 | 16 | 0 | 25 | Ref. [38] Table 4 |
| Offshore wind | 2280 | 0 | 49 | 0 | 30 | Ref. [38] Table 5 |
| Biofuel | 2300 | 0 | 94 | 6 | 20 | Ref. [38] Table 48, ref. [38] |
| Hydropower run of river | 0 | 0 | 169 | 0 | 60 | Ref. [38] Table 14 |
| Hydropower with reservoir | 0 | 0 | 101 | 0 | 60 | Ref. [38] Table 12 |
| Pumped hydro storage | 0 | 0 | 7 | 0 | 55 | Ref. [39] |
| Short term storage | 86 | 101 | 1 | 0 | 10 | Ref. [39] |
| Long term storage | 1612 | 9 | 14 | 0 | 15 | Ref. [39] |
| AC transmission^ | 900 | 0 | 0 | 0 | 60 | Ref. [38] Table 39 |

^AC transmission overnight cost is given in €/kW/1000km.

While future cost is uncertain, I am using expected values in the deterministic system design model. To cover the aspect that future cost is not known exactly, I handle cost uncertainties in the generation steps that follow the system design phase.

these cost components. I uniformly assume cost of capital to be 7.3%, which has been the historic average [37].

## 2.3 Cost uncertainty

Cost of almost all components of future renewable electricity systems can be expected to fall compared to today. Cost falls with deployment as production processes get improved, product understanding increases with the use, and financing can be provided with lower overheads. Exactly how much cost will fall with deployment is not known, however. To cover this uncertainty, I am using minimum and maximum estimates of cost [38] of the four supply technologies analysed in this study. Because I do not have any other information about how likely any cost developments are, I am following the principle of maximum entropy and I am assuming a uniform distribution between minimum and maximum estimates, see Table 2.

I only consider uncertainty in cost of on- and off-shore wind, and utility-scale and rooftop PV, first because these are the technologies whose cost-effectiveness I am assessing in this study. Second, because cost has little impact on the system design. Because I enforce supply shares in the system design model, the extent to which supply technologies are deployed is

**Table 2. Uncertain input parameters.** Parameters with uniform distribution are represented by their min and max values. Parameters with normal distribution are represented by their mean and standard deviation.

| Name | Description | Distribution | Min/Mean | Max/Std | Source |
|---|---|---|---|---|---|
| Cost onshore wind | Cost scaling factor of onshore wind. | uniform | 0.727 | 1.545 | Ref. [38] (Table 4) |
| Cost offshore wind | Cost scaling factor of offshore wind. | uniform | 0.785 | 1.434 | Ref. [38] (Table 5) |
| Cost rooftop PV | Cost scaling factor for rooftop PV. | uniform | 0.864 | 1.136 | Ref. [38] (Table 9) |
| Cost utility-scale PV | Cost scaling factor for utility-scale PV. | uniform | 0.538 | 1.115 | Ref. [38] (Table 7) |
| Land requirements wind | Onshore wind capacity density [W/m$^2$]. | normal | 8.820 | 1.980 | Ref. [8] |
| Efficiency utility-scale PV | Module efficiency of utility-scale PV. | uniform | 0.175 | 0.220 | Ref. [40] |
| Land requirements utility-scale PV | Share of land that is covered by PV modules. | uniform | 0.400 | 0.500 | Refs. [7, 41, 42] |

determined by the enforced shares, not by cost. Changes in cost of other technologies could lead to different designs, for example if hydrogen storage cost is much higher than the expected value, hydrogen may be replaced with bioenergy. For this reason, I use expected values only (see Table 1) for all other components other than the four wind and solar supply technologies.

## 2.4 Land requirements

To determine land requirements of supply technologies, I assume capacities of technologies always to require the same amount of land and therefore apply a proportional constant to installed capacities: the inverse of capacity density, given in square meters per Watt. As the range of capacity density values given in the literature is large for onshore wind and utility-scale PV, I am using a stochastic approach here as well.

Land requirements of onshore wind in this study are those of the wind turbines together with the technically necessary spacing between turbines. While the spacing can be used for agriculture, it excludes other land uses and the spacing also does not reduce visual impacts. Thus, I include spacing in the land requirements of wind turbines in this study (see Section 3 in S1 File for an analysis excluding spacing). The land requirements estimates of wind and solar power furthermore include all additional infrastructure necessary: substations, access roads, and service buildings.

Theoretically, the capacity density of onshore wind can be high: based on technical specifications, it is up to 19 W/m$^2$ for the best turbines and ~10 W/m$^2$ on average [43]. However, in deployed wind farms, the capacity density is lower, with values between 2–10 W/m$^2$ [7,8,44,45]. This can have different reasons: the capacity density depends on the placement of turbines in the wind park and it depends on the technical specifications of the wind turbine. More importantly, because land is not the most important cost component of wind farms, farms are not necessarily build in a way to maximise capacity density. In this study, I am using a capacity density estimate taken from ref. [8] based on measurements in the US. Here, the authors found 8.8 W/m$^2$ on average with a standard deviation of ~2 W/m$^2$, see Table 2.

Uncertainty about land requirements of utility-scale PV is similarly high. Around 40%–50% of the area of solar farms is covered by modules [7,41,42], while the rest of the land is used for inverters, power lines, spacing, and roads. In addition, the technology used, orientation, and efficiency of the PV modules have great impact on land requirements as well as they determine the capacity installable on the area covered by modules. Theoretically derived capacity densities using today's module efficiencies are in the range of 80–100 W/m$^2$ [42], but measurements from the US show much lower estimates of 20–30 W/m$^2$ only [7,8,44]. A recent study from Germany shows, however, that the capacity density of German utility-scale PV has increased drastically over time: in less than 20 years, it increased by factor 3 to ~70 W/m$^2$ in 2018 [46]. The authors explain the trend not only by increasing module efficiencies, but also by more economic use of land. These findings show that theoretically derived capacity densities may actually be more accurate for future projections than historic measurements, and I am therefore using capacity densities derived from theory. I am assuming that land is covered to 40–50% by modules [7,41,42], and that module efficiency is between 17.5–22% [40], see Table 2. Assuming a uniform distribution for both, this leads to an expected capacity density value of utility-scale PV of ~88 W/m$^2$.

The remaining two supply technologies, rooftop PV and offshore wind, have no land requirements whatsoever. Rooftop PV is built on existing structures, and offshore wind is not built on land. This makes them promising options for reducing total land requirements of renewable electricity systems.

Hydroelectricity, bioenergy, storage, and transmission will all require land which I am not analysing in detail in this study. The reasons why I ignore land requirements of each of the components are the following: First, I do not analyse land requirements of hydroelectricity in detail, because system designs in this study all contain the exact amount that is installed today. Thus, hydroelectricity's land requirements in all cases are the same, and equal to today's. Today's requirements are not small, however. They are dominated by the extent of water reservoirs [7], which span roughly 50,000 km$^2$ (1% of total land) in Europe [47]. Not all of the reservoirs in Europe are used for electricity generation, and even less are used exclusively for it [47], so this total number can be seen as an upper bound of the land requirements of hydroelectricity, even after dam, power house, and access roads are added.

Second, land requirements of bioenergy are very small as long as only residuals are used for fuel production. When dedicated energy crops are farmed, bioenergy has the lowest capacity density of all renewable technologies [8]. The by far largest contribution to its land requirements stems from fields for crop farming, however. Because I allow only residuals to be used for electricity generation, land requirements include the power plants only, which leads to a capacity density in the order of $10^4$ W/m$^2$ [7] and thus 2–3 orders of magnitude larger than solar and wind power. This makes bioenergy's contribution to total system land requirements insignificant and I am therefore ignoring it in this study.

Third, land requirements of electricity storage depend on the amount of electricity that must be stored. Commercial suppliers offer 1 MW / 1 MWh battery storage systems in standardised container enclosings today [48], leading to a capacity density in the order of $10^5$ W/m$^2$ and $10^5$ Wh/m$^2$. Power-wise, a capacity density of this magnitude makes the land requirements of battery storage insignificant compared to the one of solar and wind power, even if spacing, roads and further infrastructure must be added. Energy-wise, capacity density cannot be compared to solar and wind power, and the total land requirements depend on the amount of electricity that must be stored in batteries. This is equally true for hydrogen storage. Here, the energy-wise capacity density depends on how hydrogen is stored. Hydrogen has a low energy density of 3 kWh/m$^3$ if stored uncompressed at normal conditions. It can be stored underground in salt caverns, or overground in steel tanks. Capacity density is lowest if hydrogen is stored in such overground tanks in uncompressed form. Together with a conservative height of the tanks of 2 meters, this equals 6,000 Wh/m$^2$. This conservative estimation is worse than the one for battery storage. Because much more electricity is anticipated to be stored as hydrogen rather than in batteries, total land requirements of hydrogen storage may be high, if it is not stored in compressed forms, in tanks taller than 2 meters, or underground, and if large amounts must be stored. The latter is not the case for the system designs I am considering in this study (see "Storage and flexible generation require small amounts of land" in results section), and thus I am ignoring land requirements of battery storage and hydrogen storage.

Lastly, the transmission grid already has significant land requirements, which will likely increase in fully renewable systems. Currently, there are 220,000 km of AC transmission lines in the study area (based on a model of the transmission grid originally created in ref. [49] and extended in ref. [50]). With an estimated 13.5 m buffer zone on each side [51], this leads to 6,000 km$^2$ (0.12%) of land required. I am not assessing the land requirements of transmission in the fully renewable scenarios of this study because its spatial resolution, the national level, is too low to determine necessary land for transmission. Two former studies with subnational resolution indicate, however, that fully renewable electricity requires only 20% larger transmission capacity when countries are net self-sufficient [21], as they are in this study, and that a transmission system twice as large as today's or larger is likely not beneficial, even if countries are not self-sufficient [52]. The land requirements of the transmission grid in a fully renewable electricity future is therefore likely well below 0.3% of total land.

## 2.5 Stochastic model

I use technology cost and technology land requirement parameters to derive total system cost and total land requirements of solar and wind power in all 286 system designs stemming from the system design phase. I do this in two steps. First, I sample 100,000 times from the input uncertainties using Saltelli's extension of the Sobol sequence [53] to derive a sufficiently large sample set of the seven-dimensional input space. Second, for each sample and each system design I derive total land requirements of solar and wind power by multiplying their installed capacities with the inverse of their capacity density. Similarly, I derive total system cost by scaling technology cost of solar and wind from the system design with the factors given from the input sample. This leads to 100,000 output observations for each system design and therefore to ~29 million observations of pairs of cost and land requirements, which I am using to analyse cost effectiveness of different supply technologies.

## 3. Results

### 3.1 Renewable electricity supply with vastly different land requirements

Among almost all of the ~29 million observations, cost of electricity in all Europe is between 0.06 and 0.10 EUR per kWh consumed while land requirements of solar and wind power are between 0% and 3% of total European land (Fig 2). The observations show that land requirements of European electricity systems can vary by more than an order of magnitude while their cost does not exceed twice the lowest cost.

When I reduce uncertainty distributions to their expected values (their means), I find that there is a trade-off between expected land requirements of renewable electricity and its expected cost. Among all 286 system designs with different supply shares, a system with only onshore wind and utility-scale PV has the lowest expected cost of around 0.07 EUR per kWh consumed and requires ~2% of Europe's total land (~97,000 km$^2$)–an area roughly the size of Portugal (see S1 Fig in S1 File for data on the national level). Cost is minimal when both technologies contribute 50% to the total capacity of wind and solar technologies. While higher shares of utility-scale PV decrease land requirements, they also increase cost (right flanks in

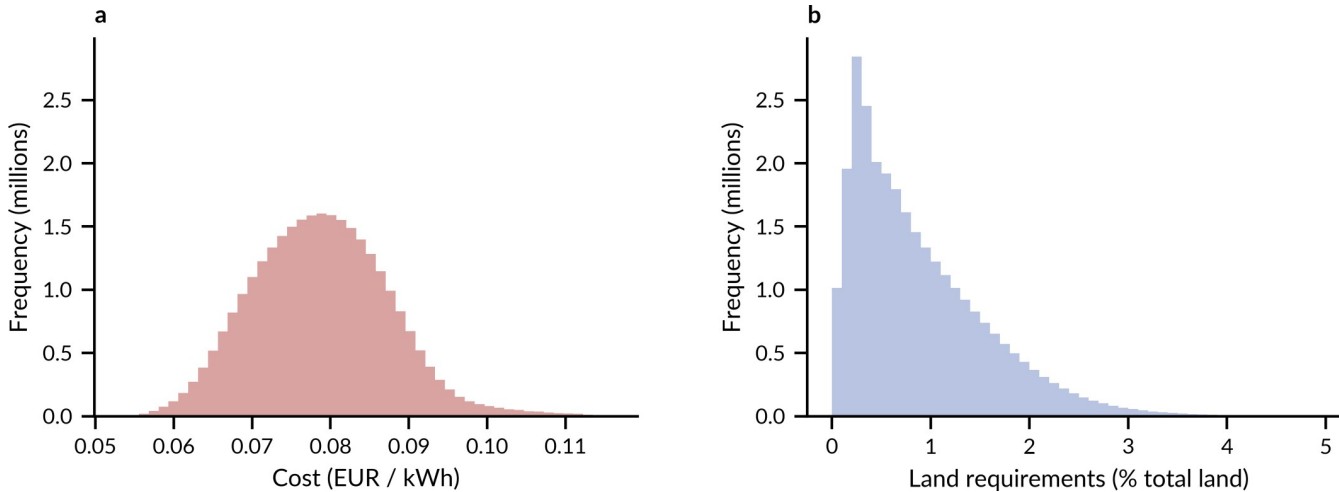

**Fig 2. All ~29 million observations of cost and land requirements of fully renewable electricity systems.** The observations include all possible supply share combinations, including systems supplied, apart from hydroelectricity and bioenergy, exclusively from onshore wind, utility-scale PV, and rooftop PV, or with high shares of offshore wind. The ranges furthermore contain technology cost and technology land requirement parameters from the full range of their uncertainty. **a,** System cost relative to electricity demand. **b,** Land requirements of solar and wind power, relative to total land in Europe.

Fig 3A and 3B). Higher shares of onshore wind increase both cost and land requirements. A system design with only onshore wind has the highest expected land requirements (top corner of Fig 3B). Rooftop PV has the largest potential to decrease land requirements, as it requires no additional land, but it also increases cost the most (left corners of Fig 3A and 3B).

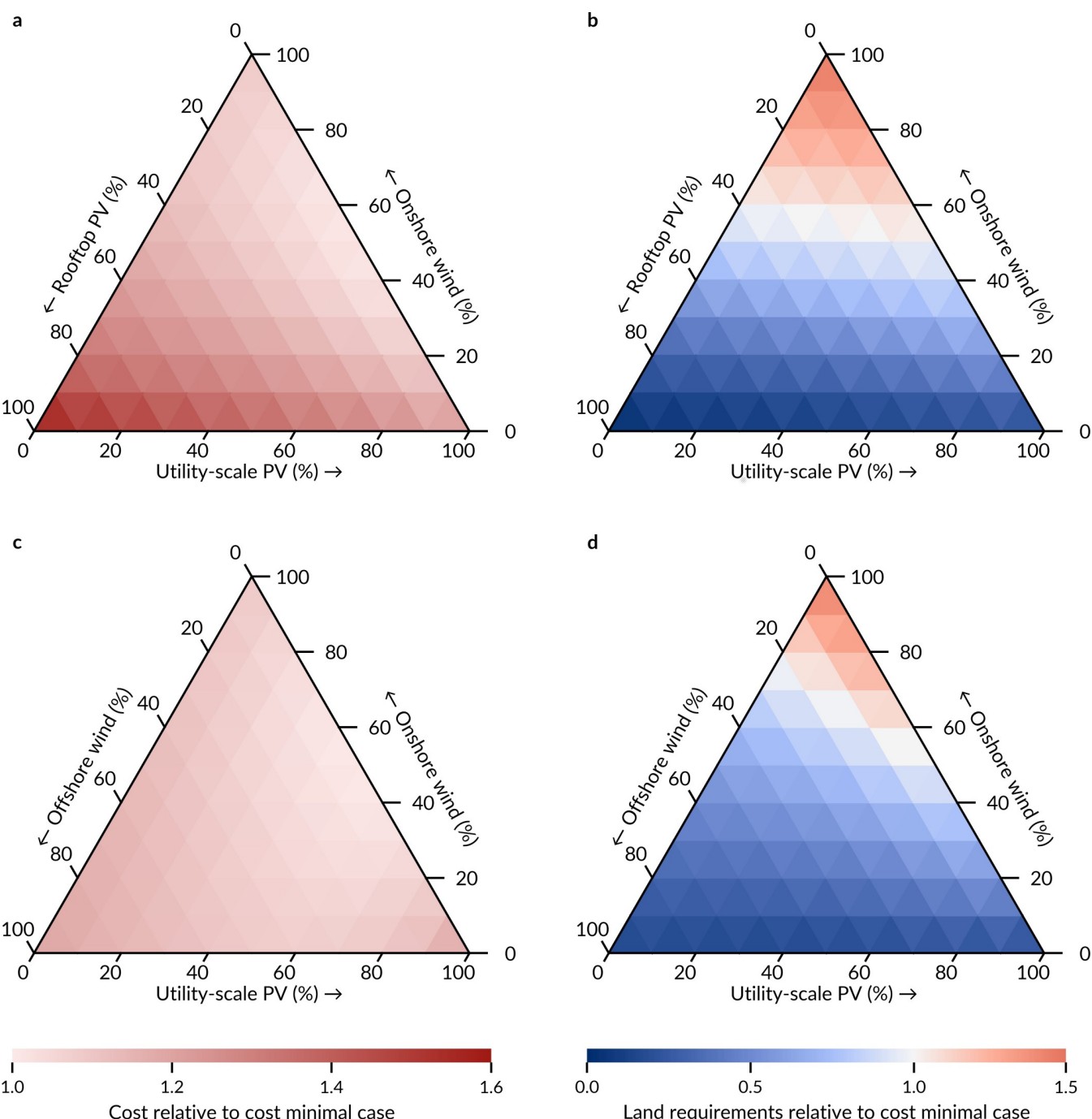

**Fig 3. Expected cost and land requirements of fully renewable electricity systems with all possible shares of three different supply technologies.** Cost and land requirements are relative to the case with minimal cost of ~0.07 EUR per kWh consumed which requires ~2% of Europe's total land. All cases include hydroelectricity of today's capacity and bioenergy from residuals next to three solar and wind technologies. Expected values are the means of uncertainty distributions. **a,b,** Total system cost (**a**) and land requirements (**b**) of cases with utility-scale PV, onshore wind and rooftop PV as supply side options. **c,d,** Total system cost (**c**) and land requirements (**d**) of cases with utility-scale PV, onshore wind, and offshore wind as supply side options.

Electricity system designs with offshore wind in addition to onshore wind and utility-scale PV have lower cost when they do not include rooftop PV (Fig 3C). The potential of offshore wind to decrease land requirements is smaller than the one of rooftop PV, but only slightly (Fig 3D). While offshore wind requires no additional land–similar to rooftop PV–it is not available in every country in Europe and is replaced by onshore wind in these places. Compared to onshore wind, the land requirements of all other three supply technologies–offshore wind, utility-scale PV, and rooftop PV–are lower and thus, system designs with large shares of any of these alternatives have smaller total land requirements, albeit at higher cost.

### 3.2 Offshore wind reduces land requirements most cost-effectively

The rather large expected land requirements of the cost-minimal case can be reduced most cost-effectively by replacing onshore wind with offshore wind. In this way, total land requirements of renewable electricity can be reduced by 50% (to 1% of total land) for a cost penalty of 5% (Fig 4A). This cost penalty corresponds to 0.22 EUR per m$^2$ and year and comes at a share of offshore wind of ~25%. Land requirements can be decreased further, in total by 85%, with higher shares of offshore wind. However, cost increase sharply for the last minor reduction in land, for which utility-scale PV must be phased out (left-most point in Fig 4A).

Reducing land requirements with utility-scale (Fig 4B) and rooftop (Fig 4C) PV has higher cost. To reach the same reduced land requirements of 50% below the cost-minimal case higher shares of utility-scale PV lead to a cost penalty of ~9%, corresponding to 0.41 EUR per m$^2$ and year. Cost rises progressively however, and early decreases of land requirements have very low cost. In total, 80% of the cost-minimal land requirements can be removed with a cost penalty of 23% using utility-scale PV only.

The highest expected cost comes with phasing-in rooftop PV. Here, reducing land requirements by one square meter requires 0.75 EUR per year when reducing cost-minimal requirements by 50% (to 1% of total land, see Fig 4C)–this is a cost penalty of 17%. Similar to utility-scale PV, the cost increases with higher rooftop shares and the largest increase can be explained by the technology that is phased-out: the first half of rooftop PV replaces onshore wind, while the second half replaces utility-scale PV at a much higher cost. The increase of

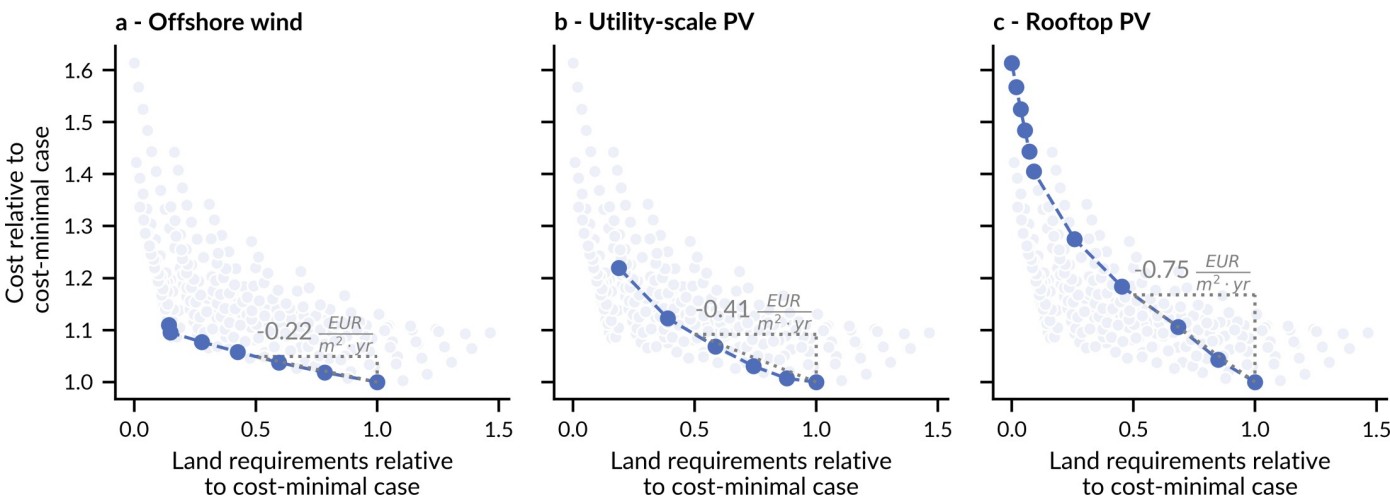

**Fig 4. Cost-effective ways to reduce expected land requirements using supply technologies individually.** All panels show expected cost and expected land requirements of all 286 system designs in light blue in the background. **a–c,** Dark blue cases show Pareto-optimal decreases of land requirements from cost-minimal case using offshore wind only (**a**), utility-scale PV only (**b**), and rooftop PV only (**c**).

offshore wind, utility-scale PV, and rooftop PV shares always reduces expected land requirements of fully renewable electricity systems, albeit at different cost.

### 3.3 Cost penalties of 20% or less are most likely even for low land requirements

Uncertainty in technology cost and land requirements leads to high uncertainty in the cost penalties for renewable electricity with lower land requirements. To ensure land requirements are below 1% of total European land, cost penalties can be as large as 40%, but are most likely below 20% for all supply technology options (Fig 5). For offshore wind and utility-scale PV a cost penalty below 20% can be expected in 75% of the cases. In a quarter of all cases, there is no cost penalty necessary at all, because the cost-optimal case has land requirements of 1% or lower. With lower thresholds, cost penalties become more likely and also larger. For a threshold of 0.5% of European land, a cost penalty of 20% is still more likely for the more cost-effective technologies offshore wind and utility-scale PV.

Uncertainty does not alter the order of cost-effectiveness of the three supply technologies with low land requirements: rooftop PV is always the least cost-effective technology. Offshore wind is most cost-effective, but only when large amounts of onshore wind are to be replaced (land area thresholds of 1% or smaller in Fig 5). In these cases, offshore wind is more cost-effective than utility-scale PV. For medium land thresholds (1.5%), expected value of cost and its distribution are nearly the same for both technologies. Above that, as long as land area is to be reduced only little, utility-scale PV is the most cost-effective option.

### 3.4 Low land requirements require low shares of onshore wind

While land needs of supply technologies are uncertain, onshore wind is in any case the technology with the highest requirements for land if spacing is included (see Section 3 in S1 File for an analysis excluding spacing). Offshore wind, utility-scale PV, and rooftop PV are

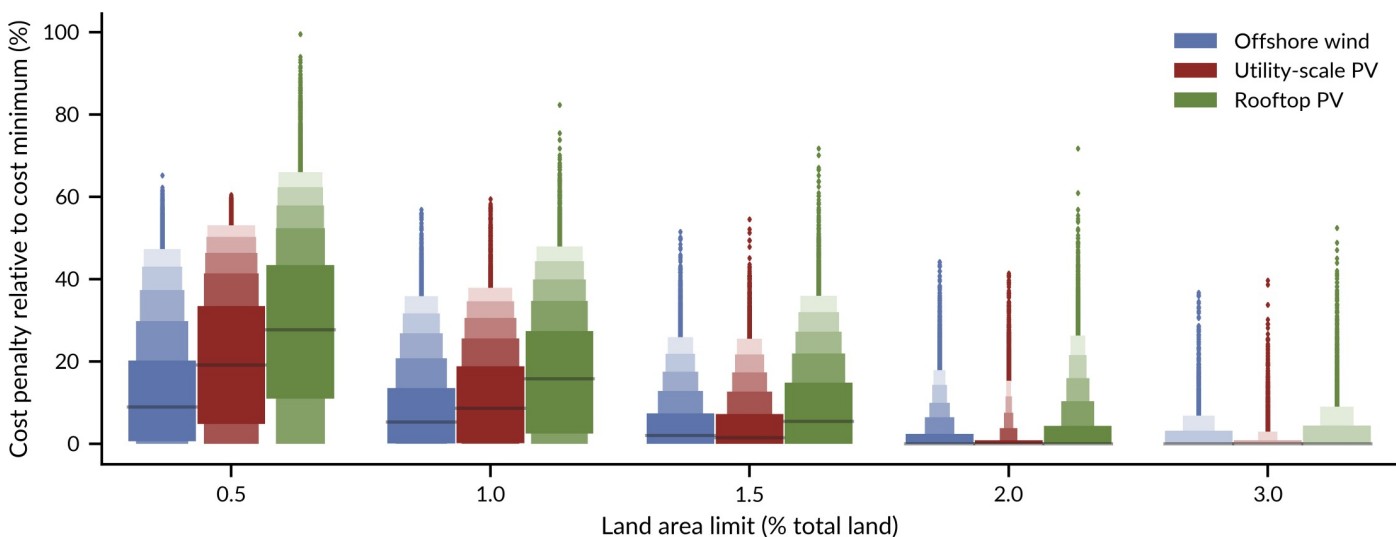

**Fig 5. Resulting cost penalties to ensure land thresholds.** Cost penalties arise from higher than cost-minimal shares of one of three supply technologies: offshore wind, utility-scale PV, or rooftop PV. The uncertainty distribution of cost penalties is displayed using letter-value plots. Letter-value plots are an extension to boxplots for large data. Dark grey lines indicate the median value of the cost penalties, and the widest boxes above and below the median visualise the 25–75% quantiles. Each following box contains half as many observations as the box closer to the median. The extreme 1% of the observations are considered outliers and marked with rhombs.

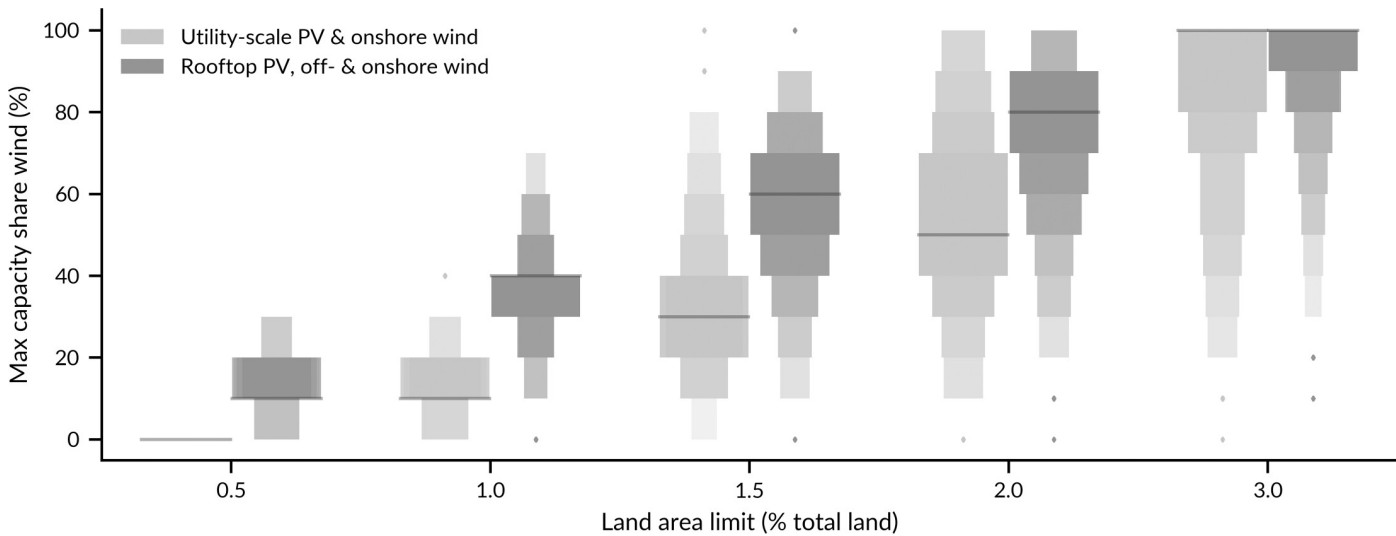

**Fig 6. Maximal capacity shares of onshore wind to ensure land thresholds.** Visualised shares are the maximum shares among system designs with only utility-scale PV or only rooftop PV and offshore wind in addition to onshore wind and given uncertainty. Uncertainty stems from the uncertainty of how much land onshore wind and utility-scale PV require. The uncertainty distribution of capacity shares is displayed using letter-value plots. Letter-value plots are an extension to boxplots for large data. Dark grey lines indicate the median value of the cost penalties, and the widest boxes above and below the median visualise the 25–75% quantiles. Each following box contains half as many observations as the box closer to the median. The extreme 1% of the observations are considered outliers and marked with rhombs.

therefore no-regret options to reduce the spatial extent of renewable electricity generation on land. In 50% of the cases, a land threshold of 1% of total European land can only be reached if the capacity share of onshore wind is 40% or lower (Fig 6), and if there are no additional land requirements from utility-scale PV. When utility-scale PV exists as well, onshore wind capacity must be even lower, and if utility-scale PV is the only alternative, onshore wind capacity must be as low as 10%. Renewable electricity with low requirements for land can only be reached by low shares of onshore wind.

## 3.5 Storage and flexible generation require small amounts of land

Systems with different shares of solar and wind capacity require different balancing capacity in terms of electricity storage, bioenergy, and transmission. Balancing needs are moderate for cases with balanced mixes of supply technologies (Fig 7). When supply is strongly biased towards one technology, flexibility needs rise, and in some cases, they rise sharply. Exclusively- or almost exclusively-solar cases require high amounts of short-term (battery) electricity storage. In extreme cases, storage capacities alone are able to fulfil the largest part of Europe's peak demand. Short-term storage capacities in these cases are combined with very high magnitudes of bioenergy capacity of up to 50% of peak demand to balance solar's seasonal fluctuations. Cases with mainly wind require much less bioenergy capacity and short-term storage capacity, but more long-term storage capacities to balance wind fluctuations between days and weeks. In addition, they require around 2.5 times larger cross-border transmission capacity than more balanced systems. While some of these numbers are very high, especially for cases with single supply technologies, there is no reason to believe these balancing capacities could not be built.

Balancing capacities require land as well and thus add to the land requirements of the entire electricity system. In its current state, the transmission grid uses less than 0.2% of total land (see Methods). For the system designs in this study I can not determine land requirements of the transmission grid, as the spatial resolution is too low to generate estimations.

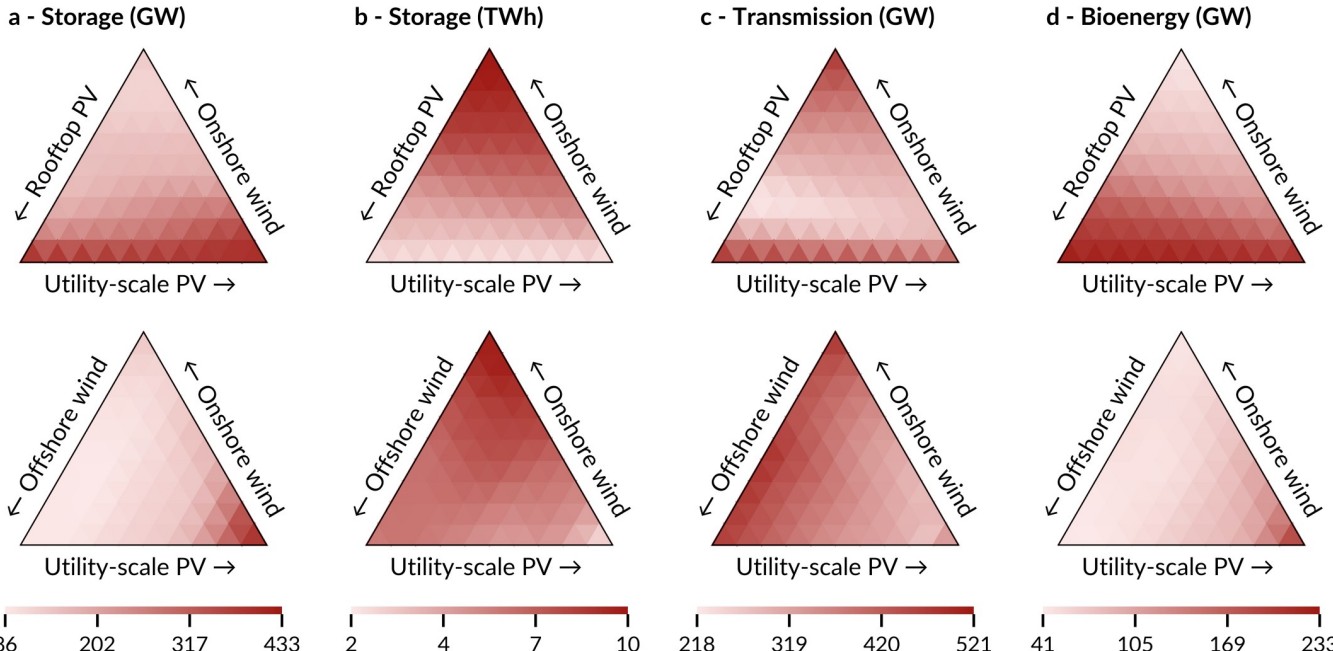

**Fig 7. Flexibility needs of fully renewable electricity systems with all possible shares of three different supply technologies.** All designs are exclusively supplied from hydroelectricity of today's capacity and different shares of three additional technologies each: onshore wind and utility-scale PV in all cases, combined with either rooftop PV (top row) or offshore wind (bottom row). Each technology is varied from 0–100% of the total capacity of the three technologies. **a–d,** Storage power capacity (**a**), storage energy capacity (**b**), cross-border transmission capacity (**c**), and bioenergy capacity (**d**). Not shown are hydroelectricity capacities which are kept constant in all cases (36 GW run of river, 103 GW / 97 TWh reservoirs, 54 GW / 1.3 TWh pumped hydro storage).

The land requirements of all other balancing technologies are very small, however. When considering $10^5$ Wh/m$^2$ for battery storage capacity, a conservative 6,000 Wh/m$^2$ for hydrogen storage capacity, and $10^4$ W/m$^2$ for bioenergy capacity (see Methods), total land requirements of all three flexibility technologies are always below 1,800 km$^2$ (0.04% of total European land). Within this estimate, the by far largest contribution comes from hydrogen, for which I use an upper-bound estimation (stored uncompressed in overground tanks). If hydrogen is stored land efficiently in underground caverns, flexibility needs of all three technologies can rise by orders of magnitude without making a significant contribution to total land requirements of fully renewable electricity systems.

## 4. Discussion

I show that there is a trade-off between land requirements and total system cost of fully renewable electricity in the future in Europe, but that reducing land requirements by changing supply-side technologies does not necessarily lead to substantial cost penalties. The expected land requirements of a system design with minimal expected cost is 97,000 km$^2$ (2% total European land). Such a low-cost system is supplied, apart from hydroelectricity and bioenergy from residuals, only from onshore wind farms and utility-scale photovoltaics (PV). Its expected land requirements can be reduced by replacing onshore wind with offshore wind, utility-scale PV, or rooftop PV. Offshore wind is the most cost-effective option of these three possibilities. It decreases cost-minimal land requirements by 50% for an expected cost penalty of only 5%. Utility-scale and rooftop PV lead to the same effect for cost penalties of 9% and 17%. All three technologies can reduce land requirements more than 50% for higher cost penalties by replacing larger amounts of onshore wind capacity.

Because future cost and land requirements of wind and solar power are not known with certainty, total system cost and total land requirements of renewable electricity supply in Europe is uncertain as well. Ensuring land requirements lower than 1% of total European land (50% of the cost-minimal case) can thus require cost penalties as large as 40%. Despite these uncertainties, three main findings are robust: First, onshore wind always requires the most amount of land and thus a switch to any other technology to reduce land is a no-regret option. Second, offshore wind is always the most cost-effective option, followed by utility-scale PV, and rooftop PV. Third, reducing land requirements of fully renewable electricity in Europe does likely not come with high cost: cost penalties of 20% or less are most likely for a system with low land requirements of 1% of European land through sufficient shares of offshore wind, utility-scale PV, or rooftop PV.

Considering all uncertainty and all possible system designs, land requirements of solar and wind power are in the range of 0–3% of total European land. Significant contributions to the land needs of electricity supply can be expected from the transmission grid and hydroelectricity (<0.3% and <1%, see Methods). The total land requirements of fully renewable electricity are thus likely within the range of 1.3–4.3%.

## 4.1 Comparison to previous studies

Comparing my results to findings from previous studies shows that there is some uncertainty about the potential of rooftop PV to reduce land requirements, as well as some uncertainty about the land requirements and therefore cost-effectiveness of utility-scale PV. There are no findings, however, that question the potential or the cost-effectiveness of offshore wind.

First, rooftop PV generates up to 1,800 TWh/yr in this study. Other estimations for the potential of photovoltaics on roofs and facades are lower: 680 TWh/yr for only rooftop PV [54] and 1200–2100 TWh for rooftop PV and PV on façades [2]. Should the lower estimations be correct, very high shares of rooftop PV as considered in this study may not be possible. In that case, rooftop PV could reduce smaller amounts of land requirements only. High shares of rooftop PV are in any case less attractive due to high cost and high balancing requirements, as I show in my results.

Second, up to 1,800 TWh/yr are generated by utility-scale PV when its capacity share is the highest. While this does not exceed potential estimations in the literature [2,28], there are conflicting estimations about how much land utility-scale requires (see also Methods). While this study uses measurements from ref. [46], ref. [55] states that areas larger than the fenced areas of PV farms must be considered, leading to capacity shares two times smaller than in this study. Whether such areas should be included is questioned [7]. Using their capacity shares would reduce the cost-effectiveness of utility-scale PV.

Third, with potential estimations as large as 40,000 TWh/yr (~10 times current electricity demand) [56] or even 50,000 TWh/yr [57], the potential of offshore wind to reduce land requirements is not questioned by previous findings in the literature.

Lastly, one study finds total land requirements of a system based on PV which are more than ten times larger than in this study: 8% of total European land [28]. The large deviation can be explained mainly by two differences: First, by the above-mentioned lower capacity densities given in ref. [55]. Second, by their finding that large overcapacities are necessary to handle renewable fluctuations: in the most extreme case of Finland, this leads to 7 times the required capacity. In my study, fluctuations are handled by continental balancing through the transmission grid and by flexible generation from bioenergy. As a result, I find only three countries require overcapacities, and overcapacities never exceed 1.15 times the required capacity. The handling of renewable fluctuations explains the largest part of the different

findings of the two studies. This shows the importance of an analysis on the system level, including not only supply but also balancing.

## 4.2 Limitations and outlook

The high-level perspective on land requirements in this study allows to understand the full spatial extent of electricity supply infrastructure on European land and its trade-off with system cost. Land requirements of the different technologies, however, are not always directly comparable. For example, while solar photovoltaics does not allow for any other land use–at least not as long as agrophotovoltaics is unavailable at large scale [58]–the vast spacing between wind turbines does allow for agriculture. Thus, the two technologies compete differently with other uses of land. In addition, building onshore wind farms on land that is already developed decreases the impact of the newly constructed infrastructure [59]. Offshore wind of course requires no land but competes with other uses of offshore areas and can have visual impacts if the turbines are close to the shore. Because I analyse total land requirements in this study, I cannot account for these qualitative differences. However, I mitigate this limitation by making options to reduce land requirements technology-specific.

Further, not only total land requirements are important, but also the exact location and technical parameters: wind turbines impact landscapes stronger when they are larger and when they are placed on exposed locations like hilltops. Analysing the visual impact of renewable electricity on landscapes has not been done on this level of detail so far. In this study, I use land requirements of solar and wind installation as a proxy for visual impacts.

By assessing the trade-off between total system cost and land requirements from a continental perspective and for fully renewable electricity supply, I omit any form of transitional or distributional aspects in this study. While this allows me to show options Europe has for its future supply, further research has to be done on the transition towards these options and on the question of how cost and land requirements are geographically and societally distributed. In particular, this includes the distributional imbalance that reducing land requirements has local impacts on land, but continental impacts on system cost. Current electricity markets cannot resolve this imbalance.

## 4.3 Conclusion

My findings show that supply technology choice is an effective way to reduce land requirements of fully renewable electricity systems in Europe. Systems with vastly different land requirements can be designed, and their cost must not vary much as long as land requirements are reduced cost-effectively. Instead of relying strongly on onshore wind, which is likely the cost-minimal solution, electricity can be generated offshore at large scale and be transported to demand centres using a sufficient transmission grid. The expansion of both, onshore wind farms and transmission grid can be limited by alternatively generating solar electricity locally. Such a solar-centred electricity supply is enabled by flexible generation from bioenergy to cope with seasonal fluctuations. These findings increase the solution space for a European energy transition and allow to integrate more diverse stakeholder positions than is possible with cost-minimised electricity system designs.

## Supporting information

**S1 File.**
(PDF)

**S1 Data.**
(NC)

**S1 Code.**
(ZIP)

# Acknowledgments

I would like to thank Johan Lilliestam and Stefan Pfenninger for valuable comments on earlier drafts of this article.

# Author Contributions

**Conceptualization:** Tim Tröndle.

**Methodology:** Tim Tröndle.

**Software:** Tim Tröndle.

**Visualization:** Tim Tröndle.

**Writing – original draft:** Tim Tröndle.

**Writing – review & editing:** Tim Tröndle.

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
