## [Decision Letter · Decision Letter 0]

29 Apr 2020

PONE-D-20-07241

Supply-side options to reduce land requirements of fully renewable electricity in Europe

PLOS ONE

Dear Mr. Tröndle,

Thank you for submitting your manuscript to PLOS ONE. After careful consideration, we feel that it has merit but does not fully meet PLOS ONE’s publication criteria as it currently stands. Therefore, we invite you to submit a revised version of the manuscript that addresses the points raised during the review process.

We recommend that it should be revised taking into account the changes requested by the reviewers. Since the requested changes includes Major Revision, the revised manuscript will undergo the next round of review by the same reviewers.

We would appreciate receiving your revised manuscript by Jun 13 2020 11:59PM. To enhance the reproducibility of your results, we recommend that if applicable you deposit your laboratory protocols in protocols.io, where a protocol can be assigned its own identifier (DOI) such that it can be cited independently in the future. For instructions see: http://journals.plos.org/plosone/s/submission-guidelines#loc-laboratory-protocols

We look forward to receiving your revised manuscript.

Kind regards,

Baogui Xin, Ph.D.

Academic Editor

PLOS ONE

Journal Requirements:

2. Please include captions for your Supporting Information files at the end of your manuscript, and update any in-text citations to match accordingly. Please see our Supporting Information guidelines for more information: http://journals.plos.org/plosone/s/supporting-information

Reviewers' comments:

Reviewer's Responses to Questions

**Comments to the Author**

1. Is the manuscript technically sound, and do the data support the conclusions?

Reviewer #1: Yes

Reviewer #2: Partly

2. Has the statistical analysis been performed appropriately and rigorously? 

Reviewer #1: No

Reviewer #2: I Don't Know

3. Have the authors made all data underlying the findings in their manuscript fully available?

Reviewer #1: Yes

Reviewer #2: Yes

4. Is the manuscript presented in an intelligible fashion and written in standard English?

Reviewer #1: Yes

Reviewer #2: Yes

5. Review Comments to the Author

Reviewer #1: This is an interesting paper covering and important topic on how multicountry/regional energy strategies impact land cover change and costs of electricity. I think the overall methodology is sound, even if done at fairly low spatial resolution. There is a time and place for these sorts of broad, generalized models, and I think the intersection of land cover change and energy production is one of them. We need this work done in the US! Thus, I’m supportive of the work being published.

My main concern with the existing manuscript boils down to parameter inputs for the model. Regarding costs, I’m not entirely sure what these costs represent and how they relate/translate to the price of electricity for consumers, which seems to be what really matters. This needs to be described in more detail instead of just citing the sources of the data. Also, the discussion section needs to tie the models results back to the real world better. I’ve seen numerous reports on the large price tag in Germany for their renewables (Energiewende). It would be nice if you could bring the model results back to what is happening in Europe/Germany with prices. You concluded prices won’t go up much, yet Germany seems to suggest your results are wrong. This needs apparent contradiction needs to be added to the discussion to make your simulations more relevant.

Regarding energy capacity. I really think you should run your simulations with and without the spacing between turbines. Energy capacity of wind is quite high when you remove spacing between turbines and simply use estimates of the actual land disturbed by the facility. As I mention in my detailed comments below, the land between turbines is used and the reality is that the existing landscape drives turbine placement, more than turbines drive land use. The point is that turbines are often placed on the landscape in areas where the existing land use can continue..so it is not wasted space. I think your results will be fundamentally different if you do this. Such a large (expected) difference in outcomes means your model is likely VERY sensitive to changes in energy capacity. The values I pulled from the literature for wind energy without the spacing, are quite high…much higher then solar.

Detailed comments.

Line 17. “that must not be large”. Replace with “with cost penalities between 5 and 10% depending on the scenario.” or just “small cost penalties”.

Line 36. Devaluate to devalue.

38. does not to do not.

61. replace must not with do not.

79. Delete ‘To be able “. Start with To identify…

168. Rooftop PV. Is this residential? If so, how is the cost calculated? I don’t know how it works in Europe, but in the US, individual households finance their rooftop systems. This can be done in many ways…from leasing the panels, the buying the fully, to taking out a loan to cover their costs. What does the 880 represent in this case? For some households, the PV reduces electricity costs and, after the loan is paid off, represents a form of tax-free income. Does the 880 represent costs to the government if they were going to provide funding for residential PV? I have further comments below (rel. to line 359( about costs. You need to add some detail here about what exactly these costs represent and why they a reasonable input for your model. For example, it seems like cost to consumers is the most ideal cost to include in your model, but perhaps this is impossible to get.

182. delete ‘in the following’

183. I think it’s good to only consider uncertainty on wind and solar while holding the other technologies constant. However, it needs to be explained better. I think you structure the model so that cost does not affect supply shares, which is fine given the goal of understanding how different energy strategies change both land cover and costs. It seems to me the main reason for doing this is to deal with the uncertainty you have in future costs of the technology. The uniform distribution is good.

192-199. The only problem with ref. 8 for an estimate of energy density is that many, if not all of the studies regarding onshore wind didn’t actual measure the area used. Instead, they estimated it. I only know of a few studies used aerial photography to directly digitize then estimate the area transformed by onshore wind. I don’t think these corrected for capacity factor. They may be worth comparing to ref. 8. Diffendorfer has a table of estimates in (ha/MW) similar to ref. 8. For example, Diffendorfer estimated an average of 0.93 ha/MW, which (if I did my conversions correctly) is 1.075 MW/ha or 107.5 W/m^2…much higher than 8.82 used in this paper. These values represent just the disturbance caused by the wind farm, not the entire project area.

Diffendorfer, J.E., and R.W. Compton. “Land Cover and Topography Affect the Land Transformation Caused by Wind Facilities.” PLoS ONE 9, no. 2 (2014): e88914.

Jones, N.F., and L. Pejchar. “Comparing the Ecological Impacts of Wind and Oil & Gas Development: A Landscape Scale Assessment.” PLoS ONE 8, no. 11 (2013): e81391

Jones and Pejchar estimated 247m2/317 MMBTu.

I’m not sure I would use “the wind turbines together with the technically necessary spacing between turbines”. The area between turbines is not useless. It can be agriculture, pasture, and habitat for many species. . You are correct that some land uses are not compatible with wind energy..homes between turbines would be very dangerous. But, at least in the US, turbines are only placed on areas where land use is compatabile…in farmlands, or pastures, or natural areas. The wind energy is not excluding the use of land…it’s exactly the opposite. Certain uses of the land excludes where wind energy goes. You could limit the potential area for wind energy in the EU by masking out urban areas, areas of building density above X/ha, certain distances to roads, areas near airports or weather radar (turbines affect radar). There are a number of GIS based wind energy potential maps out there.

Furthermore, your point about visual impacts suggests you may want to use an area even larger than the spacing between turbines because the visual impacts of wind facilities go out quite far. This would would suggest even higher land requirements.

204-206. Are these #’s truly theoretical maximums? If so, then the papers I cited above, which directly measured land use from wind facilities, seems to contradict them.

228. Given how you measure capacity density for solar, it seems like estimate for wind that only include the real land disturbance might be more comparable. In most utlility scale PV systems I know of, areas between panels are cleared and not useable..similar to the areas around the base of wind turbines.

Ultimately, you might want to consider using both the ‘footprint’ only calculation for wind as well as a large-scale estimate (as you’ve done). Other’s have done this (Denholm) and it gives a more clear picture of wind energy’s spatial impacts.

272. “Lastly, the transmission grid already has significant land requirements….”

281. Change to low spatial resolution. Higher spatial resolution = smaller units of area per pixel. 1m^2/pixel is higher resolution than entire countries. Your study is very low spatial resolution.

289. “by applying” do you mean you multiplied total energy * 1/capacity density? Please use equations when necessary to better explain your methods. I realize it was an MC approach, but the underlying process that happens at each iteration can be described. Also, how did you deal with the outputs. Did you simply develop summary statistics (means and sd) for each scenario?

299. Figure 2 is a nice summary. What is of most interest to me are those scenarios that generate both a low cost and a low land use. Thus a scatterplot of land use vs cost would be a nice panel to add. You could then describe the %’s of the energy mixes for either extreme point…lowest cost and lowest land use vs highest cost and highest land use. Just seeing the shape of joint distribution would be helpful to me as it might show trade-offs or system bounds…for example there are no cases of low land use and low cost (a pet hypothesis of mine…particularly if you want to place wind in areas with minimized environmental impacts…it will cost more).

299. I’d make the ranges be 0% and ~3% and ~0.06 and ~0.10 as the graphs show higher and lower values.

299. You don’t need the sentence “these ranges include…” nore the next sentence. You state the figure represents all 29 million cases.

305. I don’t disagree with your statement here but think some readers my wonder why 0-3% is a ‘vast’ difference in land use and others will think that a doubling of their electric bill is a HUGE increase! In my world, an energy company has a very large fight if they try to raise electricity rates by just 3%...this really hurts low income households. So, it would be good to try to put these modelled changes in context..both social (for cost) and perhaps environmental? (for land use).

311. Figure 3 would better match the text if the cost plots were in Eur per KWH and the land use plots in % of Europe’s land total. Right now the scales don’t match what’s described in the text. I can see the logic for the current ‘difference from minimum’ scale.Perhaps % change from minimum would be better here since you describe % changes in the text quite often.

Your results will change considerably if you used the estimate of energy capacity for wind energy that considers just the land used, not the space between turbines.

340. You are correct, but somewhere you should acknowledge that offshore wind also has impacts…these are not on land obviously, but there is a growing literature on impacts from offshore wind on marine ecosystems and birds. I’m not sure but offshore sites likely restrict industrial fishing? There is also visual/social and cultural issues if the turbines can be seen from shore.

359. Cost becomes important here. See my comments above about cost of PV. Are your costs estimates similar to the levelized cost of electricity (LCOE) or do they vary by who is paying for the energy? Residential PV is paid for (typically) by the homeowner, while cost of industrial wind energy could be calculated as the cost to consumers, the price it costs the wind energy company to generate X capacity, etc. The broader point being that residential PV may not have the high cost when viewed from the households perspective over the life of the panels.

529. While I’m not entirely sure what you mean by the impacts on landscapes, the following papers might be relevant:

“Geographic Context Affects the Landscape Change and Fragmentation Caused by Wind Energy Facilities [PeerJ].” Accessed April 2, 2020. https://peerj.com/articles/7129/.

Jones, N.F., L. Pejchar, and J.M. Kiesecker. “The Energy Footprint: How Oil, Natural Gas, and Wind Energy Affect Land for Biodiversity and the Flow of Ecosystem Services.” BioScience, 2015. http://bioscience.oxfordjournals.org/content/early/2015/01/22/biosci.biu224.abstract.

“Monitoring Wind Farms Occupying Grasslands Based on Remote-Sensing Data from China’s GF-2 HD Satellite—A Case Study of Jiuquan City, Gansu Province, China - ScienceDirect.” Accessed April 2, 2020. https://www.sciencedirect.com/science/article/abs/pii/S092134491630163X?via%3Dihub.

“Energy Development in Colorado’s Pawnee National Grasslands: Mapping and Measuring the Disturbance Footprint of Renewables and Non-Renewables | SpringerLink.” Accessed April 2, 2020. https://link.springer.com/article/10.1007%2Fs00267-017-0846-z.

532. How much would your results change if you could model transmission lines and how much that contributes to overall land requirements for a given energy scenario? I’m not sure if more offshore would mean more transmission, but it could.

Reviewer #2: This paper addresses a large concern with renewable energy development, land use requirements.

Since I do not have the expertise to address the economic sections of this manuscript, I will focus on the land use requirements. Therefore I will provide more high-level comments than detailed.

1) I would have like to see more about the context of the foot print and cost in European countries (e.g., impact to natural lands)

2) There has been a lot of work done on this matter in USA. Please explain why the "Energy Sprawl" work was not cited in this paper?

3) Would it have been possible to give summary regarding different countries? It seems like Europe was considered one large mass. How do these results like up with European renewable energy policies to meet future energy demands?

4) On the above note, the author assumes no new hydropower dam development. Is this realistic?

5) It seems the author included roads and transmission lines into the land use estimate for solar and wind but not hydroelectic. This does not seem valid.

6. PLOS authors have the option to publish the peer review history of their article (what does this mean?). If published, this will include your full peer review and any attached files.

Reviewer #1: No

Reviewer #2: No

---

## [Author Response · Author response to Decision Letter 0]

4 Jul 2020

# Author’s response to the review on the manuscript PONE-D-20-07241 

First submitted to

*PLOS ONE*

on 12 March 2020

Re-submitted in revised form after major revisions on 04 July 2020.

I would like to thank the editors for the consideration of my manuscript and the reviewers for their review and the valuable comments. I updated the manuscript based on the comments of the reviewers and hope that I addressed all issues to the satisfaction of the reviewers and editors.

In addition to the comments from the reviewers I did the following:

* Applied minor fixes to the model of the underlying electricity system.

* Increased the temporal resolution of the model from 4h to 3h.

Both improvements increase the fidelity of the results but had only minor impacts not the results.

In the following, I address all comments by the reviewers one by one.

---

I would like to thank the reviewers for taking the time to review my manuscript and for providing useful comments that helped to improve it. Below, I copy all comments of the reviewers in bold and add my responses underneath. 

## Reviewer #1: 

> Main comment 1: This is an interesting paper covering and important topic on how multicountry/regional energy strategies impact land cover change and costs of electricity. I think the overall methodology is sound, even if done at fairly low spatial resolution. There is a time and place for these sorts of broad, generalized models, and I think the intersection of land cover change and energy production is one of them. We need this work done in the US! Thus, I’m supportive of the work being published.

Thank you.

> Main comment 2: My main concern with the existing manuscript boils down to parameter inputs for the model. Regarding costs, I’m not entirely sure what these costs represent and how they relate/translate to the price of electricity for consumers, which seems to be what really matters. This needs to be described in more detail instead of just citing the sources of the data. Also, the discussion section needs to tie the models results back to the real world better. I’ve seen numerous reports on the large price tag in Germany for their renewables (Energiewende). It would be nice if you could bring the model results back to what is happening in Europe/Germany with prices. You concluded prices won’t go up much, yet Germany seems to suggest your results are wrong. This needs apparent contradiction needs to be added to the discussion to make your simulations more relevant.

The aspects that you are mentioning here are important to the energy transition in Europe and elsewhere. I agree that transitional and distributional (be it geographically or between different types of investors or users) effects would be very relevant and should be further researched. However, I decided to not include any form of transitional or distributional effects in this study as I think it widens its scope too much.

The price developments in Germany that you describe contain both transitional and distributional aspects. They contain transitional aspects because Germany has been investing into solar power at a point in time when solar power’s cost was still very high, knowing that cost will decrease with deployment. Prices in Germany also contain at least two forms of distributional aspects. First, had Germany not invested into solar power but rather waited for another country to do so, electricity prices in Germany would be lower today. Parts of the cost of the energy transition would have been redistributed to another country. Second, the renewable electricity levy included in electricity prices in Germany strongly depends on who is paying for them. Households do pay, some businesses do, larger businesses do not. This distributional aspect of electricity prices is driven by political decisions, not by technical or economic aspects of renewable electricity.

In this study, I am not considering any of these transitional or distributional aspects of electricity prices. Doing so would require a significant shift of the focus of this study and a different research design. Instead of prices, I am assessing total system cost of electricity (by that avoiding distributional aspects) of a fully renewable electricity system (by that avoiding transitional aspects and with it, distributional aspects between countries). This lets me understand the trade-off between total system cost of fully renewable electricity and their land requirements on a long term. The study answers the question of cost of _not_ using land for electricity generation in future electricity supply. It does not answer questions of how to reach these kind of systems (transitional aspects) or who would profit (distributional aspects). I am looking forward to reading about these aspects in further studies.

I fully agree with you that the manuscript must be transparent about these choices, and I therefore applied the following changes:

1. In several places in the main text and the abstract I made clearer that the study is about future electricity supply and therefore does not reflect current cost.

2. In several places in the main text and the abstract I qualify the vague term “cost” with “total system (cost)” in order to be more precise.

3. In the “System design model” section I added the following sentence:

“Their values are future projections and assume all technologies are deployed at large scale. **In particular, this means that I ignore any forms of transitional effects stemming from technological or financial learning.**”

4. I made the caption of Table 1 more precise:

“Assumptions on installation and maintenance cost of electricity infrastructure.”

6. I added a paragraph to the “Limitations and outlook” section:

“By assessing the trade-off between total system cost and land requirements from a continental perspective and for fully renewable electricity supply, I omit any form of transitional or distributional aspects in this study. While this allows me to show options Europe has for its future supply, further research has to be done on the transition towards these options and on the question of how cost and land requirements are geographically and societally distributed. In particular, this includes the distributional imbalance that reducing land requirements has local impacts on land, but continental impacts on system cost. Current electricity markets can not resolve this imbalance.”

> Main comment 3: Regarding energy capacity. I really think you should run your simulations with and without the spacing between turbines. Energy capacity of wind is quite high when you remove spacing between turbines and simply use estimates of the actual land disturbed by the facility. As I mention in my detailed comments below, the land between turbines is used and the reality is that the existing landscape drives turbine placement, more than turbines drive land use. The point is that turbines are often placed on the landscape in areas where the existing land use can continue..so it is not wasted space. I think your results will be fundamentally different if you do this. Such a large (expected) difference in outcomes means your model is likely VERY sensitive to changes in energy capacity. The values I pulled from the literature for wind energy without the spacing, are quite high…much higher then solar.

You are right in assuming that the study results are sensitive to the decision whether to include spacing of wind turbines into the analysis. Its biggest impact is on the usefulness of utility-scale PV: without spacing, land requirements of onshore wind are smaller than the ones of utility-scale PV and thus replacing onshore wind capacities with utility-scale PV capacities increases not decreases total land requirements.

Nevertheless, I think that including spacing into the analysis and the conclusions of this analysis is the right choice. While I agree with you that wind development is driven by landscapes and existing land uses and not the other way round, the spacing of wind power does in any case exclude other land uses, has the potential to devalue land, and is visually impacting landscapes. As these are the main impacts that I am considering in this analysis, I decided to focus on land requirements including spacing.

However, I now also show results from an analysis that does _not_ include spacing. I show these results in Section 3 of Supporting Information S1 together with a short discussion of the implications of the choice of including or excluding spacing.

> Detailed comments.

> Line 17: “that must not be large”. Replace with “with cost penalities between 5 and 10% depending on the scenario.” or just “small cost penalties”.

I changed this to “with only small cost penalties”.

> Line 36: Devaluate to devalue.

Corrected accordingly

> Line 38: does not to do not.

Corrected accordingly.

> Line 61: replace must not with do not.

Thank you for this comment. Applying this replacement would lead to a sentence that does not express what I intended to express though. Cost _can_ vary a lot, but it must not vary a lot if the system is designed right.

Therefore, I changed the sentence in the following way: “[…] cost differences between supply options can be low if designed right.”

> Line 79: Delete ‘To be able “. Start with To identify…

Changed accordingly.

> Line 168: Rooftop PV. Is this residential? If so, how is the cost calculated? I don’t know how it works in Europe, but in the US, individual households finance their rooftop systems. This can be done in many ways…from leasing the panels, the buying the fully, to taking out a loan to cover their costs. What does the 880 represent in this case? For some households, the PV reduces electricity costs and, after the loan is paid off, represents a form of tax-free income. Does the 880 represent costs to the government if they were going to provide funding for residential PV? I have further comments below (rel. to line 359( about costs. You need to add some detail here about what exactly these costs represent and why they a reasonable input for your model. For example, it seems like cost to consumers is the most ideal cost to include in your model, but perhaps this is impossible to get.

The cost in Table 1 is installation and maintenance cost of electricity infrastructure. In that respect, it does not matter who is paying for the cost. I update the caption of the Table to make this clearer.

In the entire study, I am not considering cost of electricity of consumers, but rather total system cost — the total cost of electricity the European society has to pay for. With this focus, it does not matter whether rooftop PV is residential or not, as long as this does not impact total system cost. In my model, it does not.

You are of course correct in saying that rooftop PV can reduce cost of electricity of residential households, but this is usually due to many non-technical aspects like subsidies, levies, fees, and taxes. Such subsidies, levies, fees, and taxes have no impact on total system cost which I am assessing in this study.

Please see my response to your Main Comment 2 for a longer discussion and for the changes I applied in response to your comments on cost.

> Line 182: delete ‘in the following’

Changed accordingly.

> Line 183: I think it’s good to only consider uncertainty on wind and solar while holding the other technologies constant. However, it needs to be explained better. I think you structure the model so that cost does not affect supply shares, which is fine given the goal of understanding how different energy strategies change both land cover and costs. It seems to me the main reason for doing this is to deal with the uncertainty you have in future costs of the technology. The uniform distribution is good.

I agree with you that this paragraph was not clear enough. I added a longer explanation of why cost is not determining deployed capacity:

“Second, because cost has little impact on the system design. Because I enforce supply shares in the system design model, the extent to which supply technologies are deployed is determined by the enforced shares, not by cost.”

> Lines 192-199: The only problem with ref. 8 for an estimate of energy density is that many, if not all of the studies regarding onshore wind didn’t actual measure the area used. Instead, they estimated it. I only know of a few studies used aerial photography to directly digitize then estimate the area transformed by onshore wind. I don’t think these corrected for capacity factor. They may be worth comparing to ref. 8. Diffendorfer has a table of estimates in (ha/MW) similar to ref. 8. For example, Diffendorfer estimated an average of 0.93 ha/MW, which (if I did my conversions correctly) is 1.075 MW/ha or 107.5 W/m^2…much higher than 8.82 used in this paper. These values represent just the disturbance caused by the wind farm, not the entire project area.

> Diffendorfer, J.E., and R.W. Compton. “Land Cover and Topography Affect the Land Transformation Caused by Wind Facilities.” PLoS ONE 9, no. 2 (2014): e88914.

> Jones, N.F., and L. Pejchar. “Comparing the Ecological Impacts of Wind and Oil & Gas Development: A Landscape Scale Assessment.” PLoS ONE 8, no. 11 (2013): e81391

> Jones and Pejchar estimated 247m2/317 MMBTu.

> I’m not sure I would use “the wind turbines together with the technically necessary spacing between turbines”. The area between turbines is not useless. It can be agriculture, pasture, and habitat for many species. . You are correct that some land uses are not compatible with wind energy..homes between turbines would be very dangerous. But, at least in the US, turbines are only placed on areas where land use is compatabile…in farmlands, or pastures, or natural areas. The wind energy is not excluding the use of land…it’s exactly the opposite. Certain uses of the land excludes where wind energy goes. You could limit the potential area for wind energy in the EU by masking out urban areas, areas of building density above X/ha, certain distances to roads, areas near airports or weather radar (turbines affect radar). There are a number of GIS based wind energy potential maps out there.

> Furthermore, your point about visual impacts suggests you may want to use an area even larger than the spacing between turbines because the visual impacts of wind facilities go out quite far. This would would suggest even higher land requirements.

These are important observations. I understand there are in fact three questions here:

1. Should spacing be included?

2. Are the numbers correct considering spacing?

3. Should the number be even higher because I assume visual impacts?

1. I argue that it is correct to include spacing in this analysis (but I also added results and a discussion of excluding spacing). See my response to your Main Comment 3.

2. The publications you cite find indeed much lower land requirements (higher capacity densities) than the ones I am using. While I am using ~8.8 W/m2 mean, Differndorfer et al. find 108 W/m2. However, their number represents the direct surface impact which excludes spacing. Jones et al find 143 W/m2 habitat loss (when considering a capacity factor of 30%) which I did not consider at all in this study and which, I assume, does not include spacing either.

In fact, some other studies that include spacing find capacity densities lower than the one I am using here (Miller and Keith, 2018) (Nitsch et al., 2019). The reason I chose (van Zalk and Behrens, 2018) as the main reference for this study is that it combines several sources, and that it provides an uncertainty estimate which covers lower or higher values reported in other publications.

3. Yes, to correctly measure visual impact, the areas would need to be higher than the ones I am assuming in this study. The visual impact depends on the exact location of the infrastructure, for example to account for the elevation and thus visibility. Such a detailed assessment of visual impact is unfortunately not possible when considering all of Europe and when applying an energy system model. I therefore decided for using land requirements as defined in the manuscript as a proxy for visual impacts. 

To address your comment, I amended the discussion on visual impacts in the second paragraph of the “Limitations and outlook” section. Please also see my response to your Main Comment 3 for further changes.

van Zalk, J., & Behrens, P. (2018). The spatial extent of renewable and non-renewable power generation: A review and meta-analysis of power densities and their application in the U.S. *Energy Policy*, *123* 83–91. https://doi.org/10.1016/j.enpol.2018.08.023

Miller, L. M., & Keith, D. W. (2018). Observation-based solar and wind power capacity factors and power densities. *Environmental Research Letters*, *13*(10), 104008. https://doi.org/10.1088/1748-9326/aae102

Nitsch, F., Turkovska, O., & Schmidt, J. (2019). Observation-based estimates of land availability for wind power: a case study for Czechia. *Energy, Sustainability and Society*, *9*(1), 45. https://doi.org/10.1186/s13705-019-0234-z

> Lines 204-206: Are these numbers truly theoretical maximums? If so, then the papers I cited above, which directly measured land use from wind facilities, seems to contradict them.

Please see my response to your comment on Lines 192–199. In short, I don’t think the papers contradict my numbers because they measure different quantities. However, land requirements of wind power are controversially debated and to accommodate the uncertainty originating in this debate, my study contains a global uncertainty analysis which includes values that are smaller and values that are larger than the “best estimate” I took from (van Zalk and Behrens, 2018). 

van Zalk, J., & Behrens, P. (2018). The spatial extent of renewable and non-renewable power generation: A review and meta-analysis of power densities and their application in the U.S. *Energy Policy*, *123* 83–91. https://doi.org/10.1016/j.enpol.2018.08.023

> Line 228: Given how you measure capacity density for solar, it seems like estimate for wind that only include the real land disturbance might be more comparable. In most utlility scale PV systems I know of, areas between panels are cleared and not useable..similar to the areas around the base of wind turbines. Ultimately, you might want to consider using both the ‘footprint’ only calculation for wind as well as a large-scale estimate (as you’ve done). Other’s have done this (Denholm) and it gives a more clear picture of wind energy’s spatial impacts.

For both solar and wind power I estimate the full extent of the project area that is used for electricity generation and I use this as a proxy for landscape impact.

I agree with you that other measures are possible and that they would lead to different results. See Section 3 in Supporting Information S1 for an analysis that excludes spacing in wind farms.

Please see also my responses to your Main Comment 3 and to your comment on Lines 192—199.

> Line 272: “Lastly, the transmission grid already has significant land requirements….”

Changed accordingly.

> Line 281: Change to low spatial resolution. Higher spatial resolution = smaller units of area per pixel. 1m^2/pixel is higher resolution than entire countries. Your study is very low spatial resolution.

You are absolutely right. Changed accordingly.

> Line 289: “by applying” do you mean you multiplied total energy * 1/capacity density? Please use equations when necessary to better explain your methods. I realize it was an MC approach, but the underlying process that happens at each iteration can be described. Also, how did you deal with the outputs. Did you simply develop summary statistics (means and sd) for each scenario?

Yes, your assumption is correct, as is your remark that this part is not precise enough. The MC approach is not more than what you describe: for each input sample I derive an output sample of land requirements using the equation you are giving. For cost, I do something similar: I scale cost of the system design with the input sample (scaling factor from Table 2).

Regarding your question about outputs: I do not aggregate outputs. Fig 2 shows all ~29 million samples of the outputs.

To address your comment, I made minor changes to this paragraph so that the method is described more precisely. 

> Line 299a: Figure 2 is a nice summary. What is of most interest to me are those scenarios that generate both a low cost and a low land use. Thus a scatterplot of land use vs cost would be a nice panel to add. You could then describe the %’s of the energy mixes for either extreme point…lowest cost and lowest land use vs highest cost and highest land use. Just seeing the shape of joint distribution would be helpful to me as it might show trade-offs or system bounds…for example there are no cases of low land use and low cost (a pet hypothesis of mine…particularly if you want to place wind in areas with minimized environmental impacts…it will cost more).

Your questions are indeed valid questions. In my opinion, the insights you are looking for are shown in Fig 3 and 4. Fig 4 shows the scatterplot you are looking for — even if only for expected values. The figure shows that your hypothesis is correct that there are no cases that have low land requirements and low cost at the same time — meaning that there is a trade-off. The same figure also shows low cost and low land requirements (Pareto-optimal) cases in dark blue.

A scatterplot of all cases is not possible due to the amount of points to plot (29 million). In fact, the plotting libraries I use (seaborn and matplotlib) do not even manage to generate a kernel density estimate of a dataset as large. However, I think showing the scatterplot for expected values as I do in Fig 4 is more insightful in any case. The variability in the 29 million cases has two very different sources: uncertainty about model parameters, and supply shares (which are a design choice). By freezing the former in Fig 3 and 4, the analysis can focus on the impact of supply shares and show the trade-offs and bounds you are looking for. 

> Line 299b: I’d make the ranges be 0% and ~3% and ~0.06 and ~0.10 as the graphs show higher and lower values.

You are right that this was ambiguous. I changed the beginning of the sentence to “Among almost all of the ~29 million observations, […]” to be more precise.

> Line 299c: You don’t need the sentence “these ranges include…” nore the next sentence. You state the figure represents all 29 million cases.

You are correct that this is redundant information for the attentive reader. I added the information for two reasons. First, readers may choose to skim through the article and this information should help to understand the figure. Second, I wanted to make sure the reader understands that variability in the graph does not stem from uncertainty only, but also from design choices.

To address your comment, I moved the information into the caption of Fig. 1. In this way, it’s still available for the readers that may choose to skim through the article, but it’s less disturbing for the more attentive readers.

> Line 305: I don’t disagree with your statement here but think some readers my wonder why 0-3% is a ‘vast’ difference in land use and others will think that a doubling of their electric bill is a HUGE increase! In my world, an energy company has a very large fight if they try to raise electricity rates by just 3%...this really hurts low income households. So, it would be good to try to put these modelled changes in context..both social (for cost) and perhaps environmental? (for land use).

I fully agree with you that the text should not claim that a doubling of cost is less important to Europe’s population than the ranges of land requirements I found. In fact, I do not know if a doubling in cost is more or less important than a doubling in land requirements. I am not aiming at weighing cost against land requirements in this text. I see this as a political decision, not a conclusion of scientific work.

Instead, what I was trying to say here is that the range is larger. Cost doubles in the extreme cases, but you can easily find cases in which land requirements increase by an order of magnitude (0.3%—3%) or more. To reflect this better and to address your valid remark I changed the sentence to “The observations show that land requirements of European electricity systems can vary by more than an order of magnitude while their cost does not exceed twice the lowest cost.“.

> Line 311: Figure 3 would better match the text if the cost plots were in Eur per KWH and the land use plots in % of Europe’s land total. Right now the scales don’t match what’s described in the text. I can see the logic for the current ‘difference from minimum’ scale.Perhaps % change from minimum would be better here since you describe % changes in the text quite often. Your results will change considerably if you used the estimate of energy capacity for wind energy that considers just the land used, not the space between turbines.

Thank you for spotting this. I think numbers are best given relative here as I’d like the reader to focus on the relative differences between the cases rather than absolute numbers. This is also in line with the main text which mainly focusses on relative differences. But you are right that the main text additionally states absolute numbers which cannot be found in the figure.

To correct this, I added the absolute numbers of the case to which all numbers are given relatively to the figure caption.

> Line 340: You are correct, but somewhere you should acknowledge that offshore wind also has impacts…these are not on land obviously, but there is a growing literature on impacts from offshore wind on marine ecosystems and birds. I’m not sure but offshore sites likely restrict industrial fishing? There is also visual/social and cultural issues if the turbines can be seen from shore.

You are right. The “Limitations and outlook” section now includes the following sentence:

“Offshore wind of course requires no land but competes with other uses of offshore areas and can have visual impacts if the turbines are close to the shore.”

> Line 359: Cost becomes important here. See my comments above about cost of PV. Are your costs estimates similar to the levelized cost of electricity (LCOE) or do they vary by who is paying for the energy? Residential PV is paid for (typically) by the homeowner, while cost of industrial wind energy could be calculated as the cost to consumers, the price it costs the wind energy company to generate X capacity, etc. The broader point being that residential PV may not have the high cost when viewed from the households perspective over the life of the panels.

Please see my response to your Main Comment 2 and Comment on Line 168 for a longer discussion and for the changes I applied in response to your comments on cost.

> Line 529: While I’m not entirely sure what you mean by the impacts on landscapes, the following papers might be relevant:

> “Geographic Context Affects the Landscape Change and Fragmentation Caused by Wind Energy Facilities [PeerJ].” Accessed April 2, 2020. https://peerj.com/articles/7129/.

> Jones, N.F., L. Pejchar, and J.M. Kiesecker. “The Energy Footprint: How Oil, Natural Gas, and Wind Energy Affect Land for Biodiversity and the Flow of Ecosystem Services.” BioScience, 2015. http://bioscience.oxfordjournals.org/content/early/2015/01/22/biosci.biu224.abstract.

> “Monitoring Wind Farms Occupying Grasslands Based on Remote-Sensing Data from China’s GF-2 HD Satellite—A Case Study of Jiuquan City, Gansu Province, China - ScienceDirect.” Accessed April 2, 2020. https://www.sciencedirect.com/science/article/abs/pii/S092134491630163X?via%3Dihub.

> “Energy Development in Colorado’s Pawnee National Grasslands: Mapping and Measuring the Disturbance Footprint of Renewables and Non-Renewables | SpringerLink.” Accessed April 2, 2020. https://link.springer.com/article/10.1007%2Fs00267-017-0846-z.

Thank you for pointing me to this literature which I enjoyed reading. In this work I care for the visual impacts on landscapes. In this respect, I found especially (Diffendorfer et al., 2019) relevant, which I am now referencing in this first paragraph of the “Limitations and outlook” section.

Diffendorfer, J. E., Dorning, M. A., Keen, J. R., Kramer, L. A., & Taylor, R. V. (2019). Geographic context affects the landscape change and fragmentation caused by wind energy facilities. *PeerJ*, *7*, e7129. https://doi.org/10.7717/peerj.7129

> Line 532: How much would your results change if you could model transmission lines and how much that contributes to overall land requirements for a given energy scenario? I’m not sure if more offshore would mean more transmission, but it could.

This is an interesting question. An answer backed by model results would require a model that is higher resolved in space, which would increase the computational effort of this study substantially.

To address your valid question, I did two things.

First, I made the estimation of current land requirements of the transmission grid more precise. The previous estimation was an upper bound, because it included a larger area than the one I am assessing and it included circuit length, not line length. As many lines have more than one circuit, this number is too large. I now use a different method and different data, see the last paragraph in the “Land requirements” section. The new estimation of today’s land requirements is 0.12% of total land.

Second, I use two former studies with subnational spatial resolution to give a generous upper bound for the land requirements of the transmission grid in a fully renewable electricity system of 0.3% (same paragraph in the manuscript).

I do not know how land requirements vary with supply shares, but I am confident to say that this variation will likely be in the range between 0.12%—0.3% of total land and thus small compared to the variation stemming from supply technologies.

## Reviewer #2: 

> This paper addresses a large concern with renewable energy development, land use requirements.

> 

> Since I do not have the expertise to address the economic sections of this manuscript, I will focus on the land use requirements. Therefore I will provide more high-level comments than detailed.

> Comment 1: I would have like to see more about the context of the foot print and cost in European countries (e.g., impact to natural lands)

In several responses to comments from Reviewer #1 I made changes to the manuscript that made the definition of cost and land requirements clearer. For cost, these changes are mainly in the “System design model” and “Limitations and Outlook” sections. For land requirements, these are mainly in the “Land requirements” section and in the Support Information S1.

> Comment 2: There has been a lot of work done on this matter in USA. Please explain why the “Energy Sprawl” work was not cited in this paper?

Thank you for hinting me to the energy sprawl work in the USA. I am now citing (McDonald et al., 2009) in the introduction, as I found this reference to be most related to this study and the situation in Europe.

McDonald, R. I., Fargione, J., Kiesecker, J., Miller, W. M., & Powell, J. (2009). Energy Sprawl or Energy Efficiency: Climate Policy Impacts on Natural Habitat for the United States of America. *PLoS ONE*, *4*(8), e6802. https://doi.org/10.1371/journal.pone.0006802

> Comment 3: Would it have been possible to give summary regarding different countries? It seems like Europe was considered one large mass. How do these results like up with European renewable energy policies to meet future energy demands?

I agree with you that an analysis on the national level would be very insightful, in particular for decisions to be made on the national level. I abstained from analysing the national level mainly because the study is designed in a way to be insightful for Europe as a whole in the first place. When I analyse the impact of different supply shares, I vary supply shares in all countries in Europe in parallel. One might want to analyse more complex variations in which countries have different supply shares (for example lower solar shares in the Northern countries). However, I did not include such variations and doing so would increase the space of possible solutions a lot.

I further agree with you that a study with strong links to existing national policies would be very insightful, but it would also require scenarios that are informed by these policies and therefore a different research design than mine. In my study, I focus on the solution space of the entire continent. 

To address you comment, I created a map visualising the expected land requirements of the case with minimal expected cost, see Figure 1 in S1 Supporting Information. Because all countries are connected, it is difficult to allocate cost to countries and therefore I did not create a similar map for cost.

> Comment 4: On the above note, the author assumes no new hydropower dam development. Is this realistic?

You are correct that this is a conservative assumption. Indeed, a previous study finds that European hydropower generation can be increased by 50% based on technical, economic, and ecological constraints (Gernaat et al., 2017). However, the authors estimate that 85% of this increase would come from run-of-river plants without dams and they ignore social constraints. Further, integrating their results into my model is challenging as the spatial distribution of the generation increase and the associated time series of water in-feed into the new hydro stations are not known to me. Partly because of this challenge, a no-expansion assumption for hydropower is quite common in European models, e.g. (Zappa et al., 2019), (Hörsch et al., 2018), and (Pleßmann & Blechinger, 2017).

I consider the impact of this conservative assumption to be small for my results. While larger capacities of run-of-river hydropower will decrease the amount of solar and wind power necessary, it will do so only in a very limited magnitude (from today ~10% to max 15% of total demand (Publications Office of the European Union, 2019)) and it will have largely the same impact in all 286 cases. Only the small fraction of future hydropower that can be equipped with a dam can impact my results, as more dispatchable hydropower can reduce the need for other forms of flexibility and will thus likely decrease cost especially of solar-centred cases. Because the potential for hydropower with reservoir is small, I consider its impact to be small as well.

Based on your comment, I added a discussion of this aspect to Section 1 of S1 Supporting Information and link to it from the “System design model” section in the main text.

Publications Office of the European Union. (2019). *EU energy in figures: statistical pocketbook 2019*. European Union.

Zappa, W., Junginger, M., & van den Broek, M. (2019). Is a 100% renewable European power system feasible by 2050? *Applied Energy*, *233*–*234*, 1027–1050. https://doi.org/10.1016/j.apenergy.2018.08.109

Hörsch, J., Hofmann, F., Schlachtberger, D., & Brown, T. (2018). PyPSA-Eur: An open optimisation model of the European transmission system. *Energy Strategy Reviews*, *22*, 207–215. https://doi.org/10.1016/j.esr.2018.08.012

Pleßmann, G., & Blechinger, P. (2017). How to meet EU GHG emission reduction targets? A model based decarbonization pathway for Europe’s electricity supply system until 2050. *Energy Strategy Reviews*, *15*, 19–32. https://doi.org/10.1016/j.esr.2016.11.003

Gernaat, D. E. H. J., Bogaart, P. W., Vuuren, D. P. van, Biemans, H., & Niessink, R. (2017). High-resolution assessment of global technical and economic hydropower potential. *Nature Energy*, *2*(10), 821–828. https://doi.org/10.1038/s41560-017-0006-y

> Comment 5: It seems the author included roads and transmission lines into the land use estimate for solar and wind but not hydroelectic. This does not seem valid.

It is correct that the land requirements estimates for solar and wind power include access roads. They do include transmission lines only insofar as they are used on-site to connect to the transmission grid.

The estimations of land requirements of hydro electricity are much more coarse, and this is in particular because hydro electricity is not the focus of this article. Compared to solar and wind capacity, for which large expansions of 500% or more can be expected in Europe, the expansion potential of hydro electricity is small. As I explained in my response to your Comment 4, this small expansion potential stems largely from hydro electricity without reservoir whose land requirements are small (Smil, 2015). A significant land use change due to expansion of hydro electricity is therefore unlikely in Europe.

Because reservoirs are the main driver of hydro’s land requirements, I use their extent as an estimate of the land requirements of Europe. Because by far not all reservoirs are used for electricity generation, and even less exclusively for electricity generation, I consider the estimate based on reservoirs as a conservative upper bound for the land requirements of hydro electricity in Europe.

To address your comment, I made sure these aspects are clearer explained in the article.

1. I amended the description of the land requirements of hydro electricity in section “Land requirements”.

2. I further amended the description of the land requirements of solar and wind power in the second paragraph of the same section.

Smil, V. (2015). *Power Density: A Key to Understanding Energy Sources and Uses*. The MIT Press.

---

## [Editor Report · Decision Letter 1]

17 Jul 2020

Supply-side options to reduce land requirements of fully renewable electricity in Europe

PONE-D-20-07241R1

Dear Dr. Tröndle,

We’re pleased to inform you that your manuscript has been judged scientifically suitable for publication and will be formally accepted for publication once it meets all outstanding technical requirements.

Kind regards,

Baogui Xin, Ph.D.

Academic Editor

PLOS ONE
---

## [Editor Report · Acceptance letter]

24 Jul 2020

PONE-D-20-07241R1 

Supply-side options to reduce land requirements of fully renewable electricity in Europe 

Dear Dr. Tröndle:

I'm pleased to inform you that your manuscript has been deemed suitable for publication in PLOS ONE. Congratulations! Your manuscript is now with our production department. 

Kind regards, 

on behalf of

Professor Baogui Xin 

Academic Editor

PLOS ONE